# Molecular Mechanisms of Plant Extracts in Protecting Aging Blood Vessels

**DOI:** 10.3390/nu16142357

**Published:** 2024-07-20

**Authors:** Yuxin Luo, Zeru Zhang, Weijian Zheng, Zhi Zeng, Lei Fan, Yuquan Zhao, Yixin Huang, Suizhong Cao, Shumin Yu, Liuhong Shen

**Affiliations:** 1The Key Laboratory of Animal Disease and Human Health of Sichuan Province, The Teaching Animal Hospital, College of Veterinary Medicine, Sichuan Agricultural University, Chengdu 611130, China; royuxxx77@gmail.com (Y.L.); zhangzr29@163.com (Z.Z.); zhengweijian@stu.sicau.edu.cn (W.Z.); z1319019901@126.com (Z.Z.); fanl9908@163.com (L.F.); zhaoyq2728@163.com (Y.Z.); yxhuang@sicau.edu.cn (Y.H.); suizhongcao@126.com (S.C.); yayushumin@sicau.edu.cn (S.Y.); 2Department of Pharmacology, Wuhan University School of Basic Medical Sciences, Wuhan 430071, China

**Keywords:** plant extracts, vascular aging, oxidative stress, endothelial dysfunction, cardiovascular disorders

## Abstract

Plant Extracts (PE) are natural substances extracted from plants, rich in various bioactive components. Exploring the molecular mechanisms and interactions involved in the vascular protective effects of PE is beneficial for the development of further strategies to protect aging blood vessels. For this review, the content was obtained from scientific databases such as PubMed, China National Knowledge Infrastructure (CNKI), and Google Scholar up to July 2024, using the search terms “Plant extracts”, “oxidative stress”, “vascular aging”, “endothelial dysfunction”, “ROS”, and “inflammation”. This review highlighted the effects of PE in protecting aging blood vessels. Through pathways such as scavenging reactive oxygen species, activating antioxidant signaling pathways, enhancing respiratory chain complex activity, inhibiting mitochondrial-reactive oxygen species generation, improving nitric oxide bioavailability, downregulating the secretion of inflammatory factors, and activating sirtuins 1 and Nrf2 signaling pathways, it can improve vascular structural and functional changes caused by age-related oxidative stress, mitochondrial dysfunction, and inflammation due to aging, thereby reducing the incidence of age-related cardiovascular diseases.

## 1. Introduction

Vascular senescence refers to a series of degenerative changes in vascular structure and function that occur with age, including the thickening and hardening of the vascular wall, increased arterial stiffness, loss of endothelium-dependent vasodilator function, and endothelial dysfunction [1,2], leading to vascular senescence, the pathology of which is mainly due to oxidative stress, mitochondrial dysfunction, and chronic low-grade inflammation. Aging is linked to a higher occurrence of cardiovascular and cerebrovascular diseases, including hypertension, stroke, and coronary artery disease. These conditions worsen due to alterations in blood vessel structure and function. As the population ages, the incidence of these diseases is rising rapidly, and they have become one of the major causes of death among China’s elderly population [2,3]

PE is extracted from plants using physical or chemical methods, and single or mixed biological active ingredients [4], including mainly polyphenols, polysaccharides, flavonoids, amino acids, saponins and organic acids, etc. These bioactive components are secondary metabolites of plants and have biological functions such as anti-inflammatory, antioxidant and anti-aging [5,6,7], which can improve oxidative stress, mitochondrial function, and can alleviate inflammatory response through single or synergistic effects among their active components [8,9]. In recent years, PE has been shown to extend life expectancy, promote healthy aging, and mitigate the effects of aging on the body. Research in this area has been steadily increasing, with many drugs and healthcare products now being developed to prevent aging or promote healthy aging, and these products aim to exert their antioxidant and anti-inflammatory properties to regulate cellular and biochemical processes in the body [10,11], providing protection against cardiovascular disease [12,13].

## 2. PE Mitigates Oxidative Stress in Aging Vessels

PE has antioxidant properties that modulate cellular responses, signaling pathways, and chemical mediators associated with oxidative stress processes. Since the free radical theory of aging was first proposed in the 1950s [14], a large number of studies have shown that oxidative stress is an important process in vascular aging [15,16,17]. An excessive or sustained production of reactive oxygen species (ROS), surpassing the buffering capacity of antioxidant defenses or inadequacies in antioxidant enzymes, can result in oxidative stress, which in turn affects the course of age-related cardiovascular disease [16]. ROS are partially reduced metabolites of oxygen that are generated during cellular homeostasis [18], and in the vasculature they are mainly generated by nicotinamide adenine dinucleotide phosphate (NADPH) oxidases (NOX), xanthine oxidase (XO), cyclooxygenase (COX) uncoupling, the electron transport chain (ETC)and nitric oxide synthase NOS [19]. An important source of ROS production in cardiovascular systems is NOX, and its seven isozymes are transmembrane proteins with multiple cytoplasmic and transmembrane subunits that generate superoxide anions during electron transport across membranes [20]. XO is present in endothelial cells (EC) and plasma and generates superoxide anions and other reactive oxygen products by accepting electrons from molecules of oxygen, stimulating lipoprotein receptor-1 (LOX-1) expression on macrophages and vascular smooth muscle cells and increasing ROS production [21,22]. It is the respiratory chain complexes I and III that produce ROS as superoxide anions in mitochondria, and mitochondrial dysfunction or failure of its antioxidant mechanism leads to the leakage of electrons from the ETC to react with O_2_ to form a superoxide anion. At the same time, mitochondrial-reactive oxygen species (mtROS) interfere with the function of the ETC complex through the iron–sulfur oxide center, thereby exacerbating ROS production [16,23]. NOS is a homodimeric enzyme that is uncoupled in the absence of the cofactor tetrahydrobiopterin or the substrate L-arginine, resulting in impaired production and the release of NO and an increase in highly pro-oxidative superoxide anions [24]. Oxidative stress affects vascular function through the oxidation of critical proteins or the induction of redox-sensitive transcription factors. In the event that NO is inactivated, clearance increases, production decreases, bioavailability declines, and endothelial dysfunction ensues, ultimately resulting in vascular aging [25]. NO reacts with superoxide to produce highly reactive peroxynitrite, which in excess leads to protein nitration, resulting in mitochondrial and endothelial dysfunction [26,27,28,29].

PE can scavenge ROS to alleviate oxidative stress in four ways: (1) the presence of phenolic hydroxyl groups on polyphenol molecules in PE directly scavenges ROS [30]. (2) PE activates antioxidant signaling pathways, balances cellular reactive oxygen species, improves ROS overproduction due to vascular aging, mitigates the glutathione (GSH) depletion resulting from antioxidant system impairment, and concurrently reinstates the functionality of antioxidant enzymes like superoxide dismutase (SOD) [31,32,33]; GSH functions as a nucleophilic metabolite that directly reduces ROS levels, whereas SOD inhibits peroxide anion generation, neutralizes superoxides, and hinders peroxynitrite formation and the reduction in transition metal ions [34,35]. (3) PE mediates the clearance of ROS through the regulation of nuclear factor erythroid 2-related factor-2 (Nrf2). Nrf2 is an evolutionarily conserved redox-sensitive transcription factor, demonstrating significant vasoprotective properties. Aging promotes the downregulation and dysfunction of Nrf2 levels in the vascular system, and its downregulation exacerbates oxidative stress by exacerbating the susceptibility of senescent vascular cells to ROS-mediated cellular and molecular damage [10,36,37,38]. PE enhances the DNA-binding activity of Nrf2 or upregulates its protein expression, and Nrf2 induces the expression of antioxidant enzymes such as catalase (CAT), glutathione peroxidase (GPx), and GSH. PE mediates glutathione S-transferases (GST) and heme oxygenase-1 (HO-1) so that they reduce ROS production by being expressed in conjunction with an antioxidant response element (ARE) in the promoter region of the gene, relieving the oxidative stress. Moreover, PE increased GPx expression via Nrf2, converting peroxides into their corresponding alcohols, and H_2_O_2_ mediated the monocyte chemoattractant protein-1 (MCP-1)-induced inflammatory factor expression; furthermore, the expression of vascular cellular adhesion molecule-1 (VCAM-1) was inhibited. This process helps to scavenge ROS and mitigate oxidative damage to blood vessels [32,39,40]. (4) PE has been shown to decrease the expression levels of COX-1 and COX-2 in aging blood vessels, and COX-1 and COX-2 are involved in ROS production, causing oxidative damage. Concurrently, the production of oxidized low-density lipoproteins (Ox-LDL) occurs, with the oxidation of lipoproteins serving as the primary stage in the pathogenesis of atherosclerosis, resulting in injury to EC. Both have increased expression in the vascular senescent state [41,42]. In addition to Ox-LDL, oxidative stress and inflammation are major factors in the development of atherosclerosis [43]. Furthermore, PE also resists oxidative stress by decreasing Nox1 expression [44], as well as inducing AMPK phosphorylation to significantly upregulate eNOS expression to generate NO, preventing eNOS uncoupling to generate a superoxide anion [45]. In conclusion, PE has the ability to scavenge ROS and mitigate oxidative stress by enhancing the activity of antioxidant enzymes such as GSH and SOD, modulating the Nrf2 pathway, and downregulating COX and Nox1 expression levels and other pathways.

## 3. PE Reducing Mitochondrial Dysfunction in Blood Vessels

Mitochondria are double-membrane-enveloped organelles that control energy production and apoptosis, among other cellular functions, and play a pivotal role in modulating the aging process. With age, the efficacy of the respiratory chain diminishes, leading to increased electron leakage and ROS production, and reduced cellular ATP synthesis; mtROS production is associated with age-related vascular dysfunction [46]. In aging vessels, increased mtROS has been linked to dysfunction in the electron transport chain, exacerbated by a decreased cellular GSH content and impaired Nrf2-mediated antioxidant defense responses [47,48,49,50]. Concurrently, impaired mitochondrial biogenesis in EC and smooth muscle cells in the vasculature negatively affects cellular energy, and the increased induction of mtROS production via defective electron flow through the electron transport chain leads to vascular injury [25]. It was shown that PE improved the mitochondrial respiratory chain complex’s activity, upregulated SOD, CAT, and peroxiredoxin 3 (PRDX3) expression, increased ATP generation, decreased mtROS release, and maintained mitochondrial redox homeostasis; moreover, it inhibited cytochrome c release due to mitochondrial permeabilization, thereby preventing mitochondrial-mediated apoptosis and vascular senescence [51,52,53,54]. At the same time, PE promotes the restoration of mitochondrial ATP production and inhibits the release of cytochrome c through the activation of (mitogen-activated protein kinase kinase) MEK signaling and the upregulation of the anti-apoptotic protein B cell lymphoma-2 (Bcl-2) [51]. In addition, PE stimulates peroxisome proliferator-activated receptor gamma coactivator 1 alpha (PGC-1α), regulates mitochondrial biogenesis, and inhibits mtROS generation [55]; PGC-1α is a transcription co-activator that regulates mitochondrial biogenesis and the expression of antioxidant enzymes, and the decreased expression or impaired function of PGC-1α can result in mitochondrial dysfunction and the induction of senescence [56].

PE mediates mitochondrial biosynthesis and functional improvement and in-creases expression of antioxidant enzymes through activation of sirtuins 1 (SIRT1), PGC-1α deacetylation [57]. The Sirtuins are NAD+-dependent protein deacetylases whose expression decreases with age, where SIRT1 regulates mitochondrial function in the vascular system, controlling mitochondrial biology and mtROS production, cellular energy metabolism, and the removal of damaged mitochondria through autophagy [47,58,59]. Mitophagy is a mechanism of the selective autophagy process that refers to the autophagic removal of damaged mitochondria, and plays an important role in regulating their function [60]. Whereas abnormal mitochondrial autophagy leads to mitochondrial dysfunction, the frequency of abnormalities in this autophagic process increases with age, which in turn leads to vascular dysfunction; concurrently, PE attenuates mitochondrial dysfunction by impairing ROCK1-mediated mitochondria-specific autophagy [61]. Rho-associated protein kinase (ROCK) is an effector participating in multiple cellular processes; among them, ROCK1 is a protein Ser/Thr kinase involved in the regulation of actin-myosin contraction and stabilization, apoptosis, and gene expression [62]. Moreover, Xiang et al. [61] found that PE downregulated the expression levels of Bcl-2 interacting protein 1 (Beclin1) and Ser/Thr kinase PINK1 and E3 ubiquitin ligase Parkin (PINK1/Parkin). Beclin1 and PINK1/Parkin signaling activate autophagy, and Beclin1 is a key regulator of autophagy, interacting with apoptosis pathway regulatory proteins and playing an important role in apoptosis [63]. Whereas PINK1/Parkin synergistically senses mitochondrial functional status, PINK1 accumulates on the surface of dysfunctional mitochondria while recruiting and activating the E3 ubiquitin ligase activity of Parkin, which labels damaged mitochondria for degradation via the autophagy pathway [60]. In summary, PE attenuates mitochondrial dysfunction and further delays EC senescence through the modulation of SIRT1, increasing respiratory chain complex activity, activating of PGC-1α, and downregulating Beclin1 expression.

## 4. PE Delays Endothelial Senescence

A hierarchical branching network of arteries, capillaries, and veins arrange the ECs that are functionally integrated into organs to support growth, function, and repair. The vascular endothelium is located between the blood and the vessel wall, and is a continuous single-layer screen that controls the exchange of substances between the lumen, the vessel wall, and thin-walled tissues, with the main function of producing factors such as NO, CO, prostacyclin, endothelin, and others to regulate vascular tone and vascular function, and to maintain vascular homeostasis [64,65]. In contrast, the aging of the vascular system is a pathophysiological process in which ECs undergo characteristic morphological and molecular changes. During this process, increased ROS, mitochondrial dysfunction, or DNA damage promote cellular senescence and the senescence-associated secretory phenotype (SASP) in the vascular system, promote an increased production of inflammatory cytokines and chemokines as well as alterations in the synthesis of lipid mediators, and lead to impaired vasodilatation, increased arterial stiffness, and an increased inflammatory state [66].

PE reduces the proportion of senescence-associated β-galactosidase (Sa-β-gal)-positive cells; Sa-β-gal acts as a biomarker in response to cellular senescence in vivo, and PE downregulates the expression of p16/p21/p53, all of which are established markers of senescence. A prolonged and high expression of p16 will accelerate cell cycle arrest and induce cellular senescence. While p53/p21 is a key pathway in cellular senescence, p21 is located downstream of p53, which binds to and activates the promoters of downstream target genes and participates in proliferation, apoptosis, and other processes. While p53/p21 is a key pathway in cellular senescence, p21 is located downstream of p53, which binds to and activates the promoters of downstream target genes and participates in proliferation, apoptosis and other processes. As a target gene, p21 upregulates the suppression of the activity of cell cycle protein kinases, thereby impeding gene expression and preventing cells from progressing into the S-phase of DNA replication. This ultimately leads to the inhibition of DNA synthesis and the arrest of the cell cycle [67,68]. PE also significantly increased the number of cells in the S-phase of the cell cycle and inhibited cell cycle protein D1, thereby attenuating vascular endothelial senescence [69]. Furthermore, Donato et al. [70] discovered that senescent cells will accumulate in normal arterial tissue as age progresses, and their research confirmed an increase in senescence markers in the EC extracted from the arteries and veins of healthy older adults, whose expression was negatively correlated with endothelial function. Roos et al. [71] showed that vascular endothelial function could be improved by eliminating senescent cells and p16. In addition, PE reduces senescent cell viability and decreases SASP pro-inflammatory cytokine production [6]. The expression of many SASP components is regulated by the activity of nuclear factor kappa B (NF-κB), and PE reduces the SASP expression by inhibiting the activity of NF-κB [57]. Overall, PE reduces Sa-β-gal levels, inhibits SASP, and regulates p53/p21/p16, key signaling molecules of cell cycle progression, to delay endothelial senescence and thereby control its exacerbation of inflammation, etc.

## 5. PE Inhibits Inflammation in Vascular Aging

Chronic, sterile, low-grade inflammation is a hallmark of the vascular aging process, which has important implications for age-related cardiovascular diseases such as atherosclerosis, microvascular dysfunction, and aneurysm formation [25]. Vascular endothelial cells and smooth muscle cells exhibit pro-inflammatory changes in senescent blood vessels, including the secretion of inflammatory cytokines such as interleukin 6 (IL-6), interleukin 1β (IL-1β), tumor necrosis factor alpha (TNF-α), chemokines, adhesion molecules, and matrix metalloproteinases (MMP) [25]. It was shown that PE inhibited the secretion of inflammatory factors by SASP while decreasing the levels of IL-1β, IL-6, and TNF-α. Inflammatory factors promote endothelial cell apoptosis and impair NO bioavailability through various mechanisms, including inhibiting eNOS gene expression and affecting eNOS mRNA degradation and activation, which in turn cause vascular inflammation [72,73]. Meanwhile, SASP secretes inflammatory factors while inducing inflammation in neighboring cells through paracrine secretion, which promotes the spread of tissue and organ senescence, whereas a sustained increase in inflammatory factors exacerbates extravascular matrix remodeling and atherosclerosis, accelerating plaque formation [74,75]. PE downregulates the expression of NF-κB, an important nuclear transcription factor for inflammatory cytokine expression [57], and inhibits the pro-inflammatory response induced by its signaling activation. PE also attenuates inflammatory responses by inhibiting phosphatidylinositol-3-kinase (PI3K), V-Akt murine thymoma virus oncogene homolog 1 (AKT), NF-κBp65, signal transducers and activators of transcription 3 (STAT3) protein expression, and STAT3 protein phosphorylation in the PI3K/AKT/NF-κB signaling cascade pathway [67,76]; PI3K and AKT are responsible for the activation and nuclear translocation of transcription factors (e.g., STAT, NF-κB). Furthermore, PE inhibits vascular inflammation induced by the inflammatory mediator-producing enzymes LOX-1 and intercellular adhesion molecule-1 (ICAM-1) [77]; ICAM-1 is an important cell adhesion molecule belonging to the immunoglobulin superfamily, which stimulates vascular inflammation and is activated by TNF-α expression. It also stimulates leukocyte adhesion and migration to the subendothelial region of blood vessels, leading to the recruitment of multiple monocytes to the cell membrane to form monocyte clusters and colonies. After monocyte accumulation, other inflammatory cytokines and adipocytes begin to adhere to the cell surface, leading to the narrowing of blood vessels and ultimately to vascular dysfunction [78]. Elevated blood levels of ICAM-1 can also increase the transcription of the p65 subunit of NF-κB, which exacerbate vascular inflammation [78]. PE has also been shown to significantly inhibit the expression of key enzymes of cell signaling pathways, such as COX-2 and protein kinase C. PE reduces the production of inflammatory factors such as IL-1β and TNF-α by inhibiting the expression of such enzymes [42,79], and also inhibits the JNK and p38 pathways to suppress TNF-α-induced inflammatory responses [80].

There is a complicated cross-talk between increased oxidative stress and the activation of inflammatory processes in the aging vasculature, with ROS acting as signaling molecules to activate pro-inflammatory signaling pathways such as NF-κB, which regulates the transcription of pro-inflammatory genes and enhances the expression of pro-inflammatory secretory mediators. Secondly, inflammatory factors are potent inducers of oxidative stress, such as TNF-α, which activates NADPH oxidase [81]. NF-κB activates and transcribes most of the gene targets that regulate and amplify the inflammatory response (e.g., cytokines, chemokines, apoptotic cells, and phagocytosis) while driving pro-inflammatory shifts in oxidative stress [82,83]. Studies have shown that PE upregulates the expression level of SIRT1, which interferes with pro-inflammatory molecule signaling through the inhibition of NF-κB and promotion of immunomodulatory transcription factors, and inhibits NF-κB transcription through the deacetylation of histone tails in the NF-κB promoter and of NF-κB itself [84,85,86]. Furthermore, with progressive inflammation, macrophages within the vessel wall produce more ROS, reduce NO availability, promote adhesion molecule expression, stimulate vascular smooth muscle cell hypertrophy, and activate MMP, whose upregulation induces changes in structural components of the arterial wall (e.g., decreased elastin/collagen ratio) which in turn impacts vascular remodeling, leading to reduced arterial compliance, increased stiffness, and impaired vasodilatation, which results in vascular dysfunction [24]. PE also inhibited the differentiation of THP-1 macrophages into macrophage M1, reducing ROS production and pro-inflammatory factor release, while PE-treated THP-1 macrophages released anti-inflammatory factors (e.g., IL-4, IL-10, and Arg-1) by enhancing the Nrf2/HO-1 signaling pathway [87]. In conclusion, PE inhibits vascular inflammation by decreasing the secretion of inflammatory factors (e.g., IL-1β, IL-6, and TNF-α), downregulating the expression of inflammatory mediator enzymes (e.g., LOX-1, ICAM-1), controlling the transcription of NF-κB, and mediating the release of anti-inflammatory factors from Nrf2/HO-1.

## 6. PE Improves Endothelial Dysfunction in Aging Blood Vessels

With age, a series of changes occur in blood vessels, among which ED is one of the most important clinical manifestations, being a symptom of vascular aging in the endothelial cells. ED refers to the reduction in endothelium-dependent dilatation (EDD) of endothelial cells in response to chemical or mechanical stimuli, which is manifested by functional changes in the endothelial phenotype, including the thickening of the vascular intima, alterations in the structure and function of endothelial cells, and the formation of pro-inflammatory and pro-thrombotic states [67,88]. In contrast, decreased endothelial cell capacity to synthesize and release NO, the downregulation of eNOS expression, or overproduction of ROS, impairs NO bioavailability, leading to its mediated vasodilatory function with a loss of normal vascular EDD, causing ED [89,90,91,92]. PE reduces ROS-mediated NO catabolism, upregulates eNOS expression activation to promote NO production, and increases NO bioavailability to improve vasodilation; NO, as an endothelium-derived diastolic factor, is a soluble free radical produced by eNOS and catalyzed using L-arginine as a substrate that prevents leukocyte adhesion to endothelial surfaces, as well as platelet adhesion and platelet aggregation, and inhibits the proliferation of vascular smooth muscle cells and the formation of other noncellular components that make up the matrix of the vascular wall. In addition, it has vasodilatory effects, with reduced bioavailability being a key mediator of the ED [93,94,95,96]. Simultaneous PE downregulates COX-1 and COX-2 expression levels in senescent vessels and normalizes their mediated endothelium-dependent contractile responses [97]. PE also promotes endothelial vasodilation by eliminating the ROS inhibition of EDD and reducing aging-induced vascular sclerosis by normalizing aortic wall stiffness and collagen [98]. Shahlehi et al. [99] found that PE induces vasodilation via a cholinergic pathway mediated by kaempferol 3-O-rutinoside and by muscarinic and nicotinic acetylcholine receptors present in vascular smooth muscle. In addition, PE can effectively downregulate the expression level of endothelin-1 (ET-1) and regulate the dynamic balance of active substance release from aged vascular endothelium [67]. ET-1 is an endothelium-derived constrictive factor that mediates vasoconstriction, cell proliferation and differentiation, and extracellular matrix production, and its overexpression by senescence leads to an imbalance between vasodilation and contraction, impaired EDD, and an intensification of the pro-mitotic stimulation of vascular smooth muscle cells and fibrotic processes, inducing ED [100].

Additionally, PE improves ED by modulating ion channels, attenuates the CaCl^2^-induced contraction of high K^+^-depolarizing aortic rings, and inhibits receptor-gated and voltage-dependent Ca^2+^ channels to reduce Ca^2+^ influx. This attenuates its induction of increased arterial tone and vasoconstriction, while the opening of Ca^2+^-activated K^+^ channels mediates the relaxation of the aortic rings, vasodilation, and the attenuation of vascular dysfunction [101]. In addition, eNOS uncoupling is also a key mechanism of endothelial dysfunction; in addition to oxidative stress-generated ROS leading to uncoupling, enzymatic post-translational modifications of asymmetric dimethylarginine (ADMA) and eNOS also lead to uncoupling after [102]. ADMA is a methylated arginine that reduces NO synthesis by competing for the arginine binding site of eNOS [103]. It was shown that PE promotes the phosphorylation of eNOS at Ser1177 via PI3K/Akt and induces eNOS expression in vascular aging to improve NO-mediated vasodilation [104,105]. Lee et al. [106] found that PE, in addition to reversing the d-gal-mediated decrease in eNOS serine phosphorylation, restored endothelial cell NO levels with SIRT1 expression. PE not only reversed the d-gal-mediated decrease in eNOS serine phosphorylation, but also restored endothelial NO levels with SIRT1 expression. SIRT1 affects eNOS transcription and enzymatic activity in the endothelium and regulates NO production by directly activating eNOS through deacetylation [107]; there is a positive feedback with NO, which promotes the transcription of SIRT1 [108]. Imbalance and a lack of SIRT1 regulation is a major cause of endothelial cell dysfunction and a potential mediator of age-related cardiovascular disease [109,110,111]. PE significantly increased the level of AMPK phosphorylation and its downstream gene expression [45,112], and regulates vascular senescence by activating signaling pathways such as Nrf2 and P53 through its phosphorylation of target metabolic enzymes and the regulation of the expression of related genes [113]. At the same time, SIRT1 elevates AMPK activity through the deacetylation of liver kinase B1 (LKB1), and AMPK increases SIRT1 activity by promoting NAD^+^ biosynthesis; an increase in the NAD^+^/NADH ratio can further activate SIRT1 [114,115]. In addition, PE ameliorates age-related atherosclerosis and reduces the production of advanced glycosylation end products (AGEs), while alleviating endothelial dysfunction caused by oxidative stress exacerbated by activated NADPH upon its binding to the receptor. AGEs are a heterogeneous group of bioactive molecules that are formed by nonenzymatic glycation, and their mediated cross-linking collagen and other proteins resists enzymatic degradation and interferes with collagenolysis, leading to the accumulation of collagen in the arteries with age and causing arteriosclerosis [116,117]. Atherosclerosis is defined as an increase in the velocity of the conduction wave or a change in the passive mechanical properties of the artery, resulting in a decrease in its compliance. Studies have shown that the aorta’s elasticity decreases with age, leading to impaired capacitance, an inability to properly promote blood flow and maintain diastolic pressures, etc., which ultimately affects cardiovascular disease in a number of ways [70,118]. In addition, AGEs inhibit reverse cholesterol transport by downregulating the expression of ATP-binding membrane cassette transporter proteins A1 and G1 (ABCA1 and ABCG1) on monocytes. Additionally, AGEs play a role in modifying extracellular matrix molecules through glycosylation and cross-linking alterations, thereby contributing to the advancement of atherosclerotic lesions [119,120]. Collectively, PE can induce vasodilation to restore EDD and improve endothelial dysfunction by promoting eNOS phosphorylation, increasing NO bioavailability, upregulating SIRT1 transcript expression levels, and decreasing AGEs production.

## 7. Summary and Prospects

Taken together, PE exerts a protective effect against vascular aging through the following five points; the proposed mechanism of action is shown in Figure 1 and Table 1. (1) Scavenging ROS by enhancing antioxidant enzyme activity, activating antioxidant signaling pathways, and reducing COX-1 and COX-2 expression levels to mitigate oxidative stress; (2) Increasing mitochondrial content, restoring respiratory chain complex activity, and decreasing mtROS production to maintain mitochondrial redox homeostasis and to alleviate mitochondrial dysfunction due to aging; (3) Reducing the proportion of Sa-β-gal positive cells, modulating critical cellular senescence pathways, and suppressing SASP secretion to delay endothelial senescence; (4) Regulating NF-κB, the SIRT1 pathway and the downregulation of the secretory expression of inflammatory factors to inhibit inflammation; (5) Restoring endothelium-dependent dilation, upregulating the NOS phosphorylation expression level, inducing NO production and reducing the superoxide-mediated NO catabolism to improve vascular endothelial dysfunction. However, the molecular mechanisms involved in the role of PE in protecting against vascular senescence and the interconnections between the mechanisms still need to be further investigated. Also, the efficacy of some of the PE currently referenced in this article has only been confirmed in their experiments and has not yet been confirmed in clinical applications.

## Figures and Tables

**Figure 1 nutrients-16-02357-f001:**
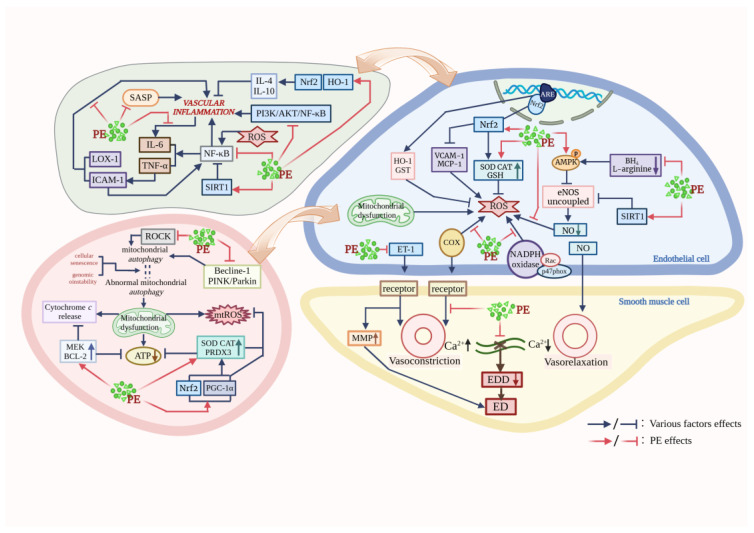
Pathways of PE protection against vascular aging. (This figure was created using BioRender).

**Table 1 nutrients-16-02357-t001:** Molecular mechanisms underlying the protective effects of PE on vascular aging.

Function	Mechanisms	Substance
Mitigates oxidative stress	Modifying cellular scavenging enzymes [31]	Aged garlic extracts (Allixin) [31]; the root extract of Vitis vinifera [121], etc.
	Activates antioxidant enzymes and upregulates the activities of glutathione, superoxide dismutase, glutathione peroxidase and catalase, inhibits lipid peroxidation [33]	The bark of M. thunbergii Sieb. et Zucc (Lignans) [33]; Quercetin [32]; Raspberry [44], etc.
	Enhances the DNA-binding activity of Nrf2 or upregulates its protein expression; conjunction with an ARE in the promoter region of the gene [10]	Verbascoside (phenylethanoid glycosides) [10]; the root extract of Vitis vinifera [121]; Cineole [122], etc.
	Induction of phosphorylated AMPK and suppressed PAI-1 expression [45]	Ginseng (ginsenoside Rb1) [45]; Ashitaba [123], etc.
	Ang II-induced senescence is attenuated through a Nox1-dependent mechanism [44]	Blackberry [44], etc.
Reducing mitochondrial dysfunction	Activation of SIRT1 and the subsequent deacetylation (activation) of PGC-1α [57]	Polyphenolic compound (Resveratrol) [57]; Rosmarinic acid [124], etc.
	Inhibits the release of cytochrome c through the activation of (mitogen-activated protein kinase) MEK signaling and upregulation of the anti-apoptotic protein B cell lymphoma-2 (Bcl-2) [51,125]	Astaxanthin (xanthophyll subclass of carotenoids) [51]; Piceatannol [125]; Trihydroxyflavone [122], etc.
Delays endothelial senescence	Downregulates the expression levels of Bcl-2 interacting protein 1 (Beclin1) and Ser/Thr kinase PINK1 and E3 ubiquitin ligase Parkin [61]	Salvianolic acid B [61]; Puerarin [126], etc.
	Downregulates the expression of p16/p21/p53 [67,68]	Canthaxanthin [67]; HuangQin [68]; Panax ginseng Meyer [127]; Artesunate [128], etc.
	Induce senescent-cell death through caspase-3 [6]	Gingerenone A [6]; Terpenoids [129], etc.
Inhibits inflammation	Inhibition of the activity of the Cytosolic Ca^2+^-dependent phospholipase A2 (cPLA2) that reduces the release of arachidonic acid [130]	Verbascoside [10], etc.
	Reduces Nox4 and p22phox expression in response to TNFα [131]	Extra Virgin Olive Oil (Luteolin) [57], etc.
	Suppresses the NF-κB pathway [80,116,125,132,133]	Ginseng (Ginsenoside Rb1) [80]; Verbascoside [132]; Curcuma longa (Curcumin) [116]; Piceatannol [125], etc.
	Decreasing expressions of PI3K, AKT, NF-κB p65, and STAT3 protein in the PI3K/AKT/NF-κB pathway and the inhibition of proteins phosphorylation [76]	Rheum palmatum L. (Rhubarb) [76]; Trihydroxyflavone [122], etc.
Improves endothelial dysfunction	Induces relaxation via muscarinic and nicotinic acetylcholine receptors present in vascular smooth muscles [99]	Gynura procumbens (kaempferol 3-O-rutinoside) [99], etc.
	Induces comparable phosphorylation of eNOS and upstream signaling kinases [95]	Tea (Flavonoid) [95]; Astragalus (flavonoids) [134]; Carthami flos [135]; Lonicerae japonicae flos (chlorogenic acid) [136], etc.
	Restores endothelial NO levels with SIRT1 expression [106]	Anthocyanins [106]; Quercetin [137]; Curcumin [137]; Resveratrol [138]; Honokiol [139], etc.
	Increases the phosphorylation level of AMPK and its downstream gene expression, targets metabolic enzymes through its phosphorylation and regulates the expression of related genes [45,113]	Ginseng (Ginsenoside Rb1) [45]; Flavonoids (Quercetin) [140], etc.
	Normalization of collagen I deposition and AGEs [26]	Curcuma longa (Curcumin) [116]; Cineole [141]; Green tea (Epigallocatechin-3-gallate) [142]; Piceatannol [143], etc.

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
