# Peer review of "Molecular Mechanisms of Plant Extracts in Protecting Aging Blood Vessels"

_nutrients, 2024, doi:10.3390/nu16142357_

Round 1

Reviewer 1 Report

Comments and Suggestions for Authors

Review of « Molecular Mechanisms of Plant Extracts in Protecting Aging Blood Vessels» submitted by Luo YX et al.

This article focused on the effect of plant extracts in protecting aging blood vessels. Aging is the originality at the vessel level. Oxidative stress OX, mitochondria M and inflammation I in the endothelial cells (EC) were developped.  Cardiovascular disorders with coronary and cerebral artery diseases, Five chapters were developped to document molecular mechanisms principaly at the endothial cell level.

The both short introduction and summary and prospects should be improved and completed.

Vascular senescence concerns also veins. The word vein is cited one fold (line 171), the same for the blood brain barrier with particular EC.

The excellent review by Donato et al (Circ. Res. 2019)  should be introduced such as «readers is directed to a recent review expanding on the aera of vein research ». This review is very complete.

Healthy aging and longevity promotion is well developped in the review of Salehi B et al (Appl. Sci. 2020). This distinction is not presented in the present review.

Lignans and tanins are compounds not presented in the present manuscript.

Autophagy is a mechanism resulting from the genomic oinstability and cellular senescence (EC and VSM from the fig. 1 should be completed in term of existing knowledge or future propects.

The healthy aging is an objective for the PE. There is not presented in the introduction.

Exemple of PE are known such as EGb 761 from Gingko biloba is used as a drug in Europe for the vascular health (Pierre et al.. 1999, 2002, Tian et al 2017). These examples of PE application should be mentionned.

Metabolites from plants that exert plant protection are used by human for improving their health during senescence (Gingko biloba,  Camellia Sinensis etc. are examples).

Visual (age-macular degeneration) and cognitive impairemts (Alzheimer) should be discussed may be as prospects with mechanism of action on DNA or epigenetic aging.

Plants extracts are known to change the gut microbiote Tea and may offer protection from fungal or bacterial pathogens (Camellia sinensis) is a good example.

A table illustrating these possible aplications should be done. Exemple for the  EC, the review from Ahmad et al. 2013 for different PE is of interest.

For the cardiac protection with agged vessels, atherosclerois is a vicious cycle of oxidative stree and inflammation involving circulating lipids, glucose. Human endothelial cells, Na,K-ATPase is inhibited by cholesterol and omega-3 fatty acids, LDL-ox.

Life style with a non sedentary aging is interesting to incorporate in this review.

May be limited data or their uncertainties (experimental, clinical) from the litterature should be underlined.

For the fig. 1, the homeostasis and sodium and potassium for vascular relaxation  should be discussed, the Na, K-ATPase is reduced in aging in addition to that of Ca2+. Furthermore a well known plant extract is the exemple of digitalis (a specific inhibitor of the Na+ pump). For digitalis which is a toxic plant by oral route, the human in agged people is finally used as a drug. 

I hope that these suggestions will improve the manuscript and interest for the readers.

A searching methodoly is presented in the abstract but not developped after. Why ?

References

Donato AJ, Machin DR, Lesniewski LA. Mechanisms of Dysfunction in the Aging Vasculature and Role in Age-Related Disease. Circ Res. 2018 Sep 14;123(7):825-848. doi: 10.1161/CIRCRESAHA.118.312563. PMID: 30355078; PMCID: PMC6207260.

Tian J, Liu Y, Chen K. Ginkgo biloba Extract in Vascular Protection: Molecular Mechanisms and Clinical Applications. Curr Vasc Pharmacol. 2017;15(6):532-548. doi: 10.2174/1570161115666170713095545. PMID: 28707602.

Pierre SV, Lesnik P, Moreau M, Bonello L, Droy-Lefaix MT, Sennoune S, Duran MJ, Pressley TA, Sampol J, Chapman J, Maixent JM. The standardized Ginkgo biloba extract Egb-761 protects vascular endothelium exposed to oxidized low density lipoproteins. Cell Mol Biol (Noisy-le-grand). 2008 Oct 26;54 Suppl:OL1032-42. PMID: 18954552.

Pierre S, Jamme I, Droy-Lefaix MT, Nouvelot A, Maixent JM. Ginkgo biloba extract (EGb 761) protects Na,K-ATPase activity during cerebral ischemia in mice. Neuroreport. 1999 Jan 18;10(1):47-51. doi: 10.1097/00001756-199901180-00009. PMID: 10094131.

Plant extracts tea

Ahmad A, Khan RM, Alkharfy KM. Effects of selected bioactive natural products on the vascular endothelium. J Cardiovasc Pharmacol. 2013 Aug;62(2):111-21. doi: 10.1097/FJC.0b013e3182927e47. PMID: 23599064.

Reviewer 2 Report

Comments and Suggestions for Authors

The authors summarized the antioxidative and anti-inflammatory effects of plant extract on vascular cells. In my opinion, since plant-derived extracts vary depending on the plant species, parts, and extraction methods, not all extracts exhibits the biological activities described in this review uniformly. This paper lacks any aspects of chemical biology and could potentially lead to significant misunderstandings among readers. Polyphenols do not all exhibit identical biological activities. For instance, they must be categorized by the derived plants or the specific compounds constituting the extract.

Comments on the Quality of English Language

Moderate editing of English language required.

Reviewer 3 Report

Comments and Suggestions for Authors

This study reviewed the effects of Plant Extracts in protecting aging blood vessels by describing the main mechanisms involved.

The study is interesting. However, there are a couple of concerns that need to be addressed:

First, The authors refer generically to plant extracts without reporting any example or related molecular mechanism of plant and natural extract.

This has also resulted in paragraphs being separated too sharply and, instead, some plants and natural extracts may perform more than one function, or different functions may be performed by different mechanisms achieving multiple effects (e.g. SIRT1 Signaling Is Involved in the Vascular Improvement Induced by Moringa Oleifera Seeds during Aging. Pharmaceuticals (Basel). 2023 May 18;16(5):761. doi: 10.3390/ph16050761. Effects of exercise training and resveratrol on vascular health in aging. Free Radic Biol Med. 2016 Sep;98:165-176. doi: 10.1016/j.freeradbiomed.2016.03.037.)

Another issue is the lack of translational medicine suggestions. the study would improve if some consideration were given to the potential of plant extracts for treating cardiovascular disease or preventing iatrogenic damage (Potential therapeutic effects of curcumin, the anti-inflammatory agent, against neurodegenerative, cardiovascular, pulmonary, metabolic, autoimmune and neoplastic diseases. Int J Biochem Cell Biol. 2009 Jan;41(1):40-59. doi: 10.1016/j.biocel.2008.06.010.      Molecules and Mechanisms to Overcome Oxidative Stress Inducing Cardiovascular Disease in Cancer Patients. Life (Basel). 2021 Jan 30;11(2):105. doi: 10.3390/life11020105.).

A table reporting the main results related to PE and mechanisms could be helpful.

Comments on the Quality of English Language

 Minor editing of English language required

Author Response

Please see the attachmen. 

Round 2

Reviewer 1 Report

Comments and Suggestions for Authors

The manuscript has been improved after the point by point reviewing. All the points have been answered. 

Reviewer 2 Report

Comments and Suggestions for Authors

The quality of the manuscript has greatly improved with more details and clarifications. I recommend the paper for publication in its present form. 

Reviewer 3 Report

Comments and Suggestions for Authors

The authors answered correctly.